# Sexually transmitted infections and prior antibiotic use as important causes for negative urine cultures among adults presenting with urinary tract infection symptoms to primary care clinics in Zimbabwe: a cross-sectional study

Ioana D Olaru [1,2] Mutsawashe Chisenga,[2] Shunmay Yeung [1,3] David Mabey,[1] Michael Marks [1] Prosper Chonzi,[4] Kudzai PE Masunda,[4] Anna Machiha,[5] Rashida A Ferrand,[1,2] Katharina Kranzer[1,2,6]

For numbered affiliations see end of article.

**Correspondence to**
Dr Ioana D Olaru;
ioana-diana.olaru@lshtm.ac.uk

## ABSTRACT

**Objective** Urinary tract infections (UTIs) are common in primary care. The yield of urine cultures in patients with UTI symptoms can be considerably different between high-income and low-income settings. This study aimed to explore possible causes of negative urine cultures in patients presenting with symptoms of UTI to primary health clinics in Harare.

**Design** Cross-sectional study.

**Setting** Nine primary health clinics in Harare, Zimbabwe.

**Participants** Adults presenting with symptoms of UTIs between March and July 2020.

**Primary outcome measures** Urine samples underwent dipstick testing, microscopy, culture, and testing for sexually transmitted infections (STIs) using GeneXpert and for the presence of antibiotic residues using an antibiotic bioassay. The primary outcomes were the number and proportion of participants with evidence of STIs, prior antibiotic exposure, leucocyturia and UTIs.

**Results** The study included 425 participants with a median age of 37.3 years, of whom 275 (64.7%) were women. Leucocyturia was detected in 130 (30.6%, 95% CI 26.2% to 35.2%) participants, and 96 (22.6%, 95% CI 18.7% to 26.9%) had a positive urine culture for a uropathogen. *Chlamydia trachomatis*, *Neisseria gonorrhoeae* and *Trichomonas vaginalis* were detected in 43/425 (10.1%, 95% CI 7.4% to 13.4%), 37/425 (8.7%, 95% CI 6.2% to 11.8%) and 14/175 (8.0%, 95% CI 4.4% to 13.1%) participants, respectively. Overall, 89 (20.9%, 95% CI 17.2% to 25.1%) participants reported either having taken prior antibiotics or having had a positive urine bioassay. In 170 (40.0%, 95% CI 35.3% to 44.8%) participants, all of the tests that were performed were negative.

**Conclusions** This study found a high prevalence of STIs and evidence of prior antimicrobial use as possible explanations for the low proportion of positive urine cultures.

## Strengths and limitations of this study

► This is the first study exploring the causes for negative urine cultures among adults presenting with urinary tract infection (UTI) symptoms to primary care in a low-income setting.

► Antibiotic exposure was assessed using two different methods.

► Testing for sexually transmitted infections (STIs) was limited to three major pathogens, which may have led to an underestimation of the prevalence of STIs.

► The prevalence of STIs is not generalisable to all primary healthcare attendees, as the participants were enrolled if they reported the presence of UTI symptoms.

## INTRODUCTION

Urinary tract infections (UTIs) are a common reason for presentation to primary care with 10% of adult women experiencing at least one episode per year.[1] Dysuria, which is one of the most common symptoms suggestive of a UTI, is reported in 2%–5% of presentations in general practice.[2] While dysuria and other urinary symptoms such as frequency and urgency may be attributed to acute bacterial cystitis, they are by no means specific.[3] Symptoms can be due to a wide range of infectious agents as well as to other non-infectious causes (table 1). In high-income settings, acute bacterial cystitis is the most common cause of UTI symptoms in women presenting to primary care.[2] Thus, the symptom complex is treated empirically with antibiotics, and diagnostic tests are not deemed routinely necessary. Patients whose symptoms do not respond to first-line antibiotics are reassessed

**Table 1** Causes of symptoms with acute onset that may be suggestive of a urinary tract infection in adults

| Category | | Specific cause |
| --- | --- | --- |
| Infectious | Acute bacterial cystitis | *Escherichia coli* |
| | | *Klebsiella pneumoniae* |
| | | *Staphylococcus saprophyticus* |
| | | *Enterococcus* spp |
| | | Unidentified (partially treated) |
| | Prostatitis | *E. coli* |
| | | *K. pneumoniae* |
| | | *Proteus mirabilis* |
| | | *Pseudomonas aeruginosa* |
| | | *Enterococcus* spp |
| | Sexually transmitted infections | *Chlamydia trachomatis* |
| | | *Neisseria gonorrhoeae* |
| | | *Trichomonas vaginalis* |
| | | *Mycoplasma hominis* |
| | | *Ureaplasma urealyticum* |
| | | *Herpes simplex* |
| | | *Candida* spp |
| | Other infections | *Mycobacterium tuberculosis* (urogenital) |
| | | *Schistosoma haematobium* |
| Inflammatory | Dermatological | Irritant or contact dermatitis |
| | | Stevens-Johnson syndrome |
| | Non-infectious | Foreign body (eg, stone) |
| | | Reactive arthritis with urethritis |
| Non-inflammatory | Drug or food related | Spermicides, topical hygiene products, drugs, certain foods |
| | Traumatic | Urogenital instrumentation/surgery |
| | | Horseback or bicycle riding |

for other diagnoses or infections with organisms resistant to first-line antibiotics.[4]

The prevalence of positive urine cultures in patients presenting with presumed UTIs varies widely across different settings. This is particularly true when comparing studies from high-income and low-income settings.[2 5] Possible explanations include (1) differences in prevalence of other conditions causing similar symptoms, (2) unregulated access to antibiotics leading to partially treated infections or (3) differences in laboratory techniques affecting the limit of detection.

This study aimed to evaluate possible causes of negative urine cultures in patients presenting with symptoms of UTI to primary health clinics in Harare.

## METHODS

Adults (aged 18 years and above) presenting with symptoms of UTIs to nine primary health clinics in Harare were enrolled into the Antimicrobial Resistance in Gram-negative Bacteria from Urinary Specimens (ARGUS) study. In Zimbabwe, primary health clinics are nurse-led and serve the population living in their catchment area. Patients accessing the acute medical services have to pay a user fee. Access to diagnostic tests is limited beyond HIV, tuberculosis and malaria. The eligibility criteria and enrolment procedures for the ARGUS study were previously described.[6] Patients were included in the study if

they presented with at least two symptoms suggestive of UTI (dysuria, urgency, frequency, suprapubic pain and/or flank pain).

This is a cross-sectional study reporting on a subset of participants from the ARGUS study who were consecutively enrolled between March and July 2020. Mid-stream urine samples were collected from study participants and underwent dipstick testing, microscopy and culture using conventional microbiology methods.[6] Leucocyturia was considered to be present if the sample had ≥$10^6$ white blood cells/L on microscopy or a positive dipstick for leucocytes. Antibacterial activity of urine indicating possible prior antibiotic exposure was ascertained using a urine bioassay. Briefly, 20 µL urine was inoculated on a 6 mm filter paper disc that was incubated on Mueller-Hinton agar and using *Escherichia coli* (ATCC25922) as an indicator organism.[7] Antibacterial activity was considered to be present in the sample if a growth inhibition zone of any size around the disc was observed following incubation for 24 hours at 37°C.

Urine samples were tested for sexually transmitted infections (STIs), specifically *Chlamydia trachomatis* and *Neisseria gonorrhoeae*. For women only, urine samples were also tested for *Trichomonas vaginalis*. STI testing was done using the GeneXpert platform (Cepheid, Sunnyvale, California, USA) according to the manufacturer's instructions. Treatment for STIs was prescribed during the initial

visit by the clinic nurse according to routine practice following the Zimbabwean national guidelines[8] and independent of study procedures. Participants with positive tests were notified of their result and offered treatment for themselves and their partners if they were not treated on their initial appointment. To ascertain if symptoms had resolved, a follow-up phone call was conducted 28 days postenrolment.

## Statistical analysis

Data were analysed using Stata V.15. Categorical data were described using counts and percentages, and continuous data were described using medians and IQRs. Differences between categorical and continuous variables were evaluated using the $\chi^2$ and Mann-Whitney U tests, respectively. The level of significance was set at a p value of ≤0.05.

## Patient and public involvement

There was no patient and public involvement in the study design. Results from this study were disseminated to nurses working in the polyclinics from where the study participants were recruited in order to improve patient care.

## RESULTS

The study included 425 participants with a median age of 37.3 years (IQR 26.8–48.0), of whom 275 (64.7%) were women and 172 (40.5%) were HIV-positive. All samples were tested for the presence of leucocytes, underwent conventional urine culture and were tested for *C. trachomatis* and *N. gonorrhoeae*. Samples from 175 women were tested for *T. vaginalis*. Of 425 urine samples collected, 420 were stored for the antibiotic bioassay; volume was insufficient for storage for the remaining 5 samples. In 255 (60.0%) of the participants, at least one of the tests was positive, while in 170 (40.0%) participants, all of the tests that were performed were negative, and the cause of their symptoms could not be determined or suspected. Figure 1 shows the distribution of positive test results (STI positive 83 (19.6%), urine culture positive 96 (22.6%), leucocyturia 130 (30.6%) and prior antibiotics 89 (20.9%)). Haematuria was detected in two (0.5%) participants.

Urine cultures were positive in 96 (22.6%), negative in 314 (73.9%) and contaminated in 15 (3.5%) participants. A higher proportion of urine cultures were positive in women (n=73, 26.6%) compared with men (n=23,

15.3%). Urine culture positivity was higher in participants with more recent symptom onset prior to presentation (58/225, 25.8% in those with a symptom onset of ≤7 days, vs 37/197, 18.8% for those with symptoms >7 days, p=0.086).

Overall, 83/425 (19.5%, 95% CI 15.9% to 23.6%) participants had an STI including 11 who also had a positive urine culture. The prevalence of *C. trachomatis* was 43/425 (10.1%, 95% CI 7.4% to 13.4%); that of *N. gonorrhoeae* was 37/425 (8.7%, 95% CI 6.2% to 11.8%); and that of *T. vaginalis* was 14/175 (8.0%, 95% CI 4.4% to 13.1%). Eleven (2.6%) participants had a positive STI test result for two pathogens. The prevalence of *C. trachomatis* was 26/275 (9.5%, 95% CI 6.3% to 13.5%) among women and 17/133 (12.8%, 95% CI 7.6% to 19.7%) among men. For *N. gonorrhoeae*, prevalence was 18/275 (6.5%, 95% CI 3.6% to 10.1%) among women and 19/150 (12.7%, 95% CI 7.8% to 19.1%, p=0.032) among men. STIs were more common in those with recent symptom onset (52/225, 23.1% among those with an onset of symptoms of ≤7 days, and 31/197, 15.7% for those with symptoms >7 days, p=0.057). Participants who tested positive for an STI were younger and were less likely to have HIV infection than those without an STI (table 2).

The median duration of symptoms was 7 days for participants with positive urine cultures and those who had an STI. Among participants who did not have an STI or a positive urine culture, the median duration of symptoms was 8 days (IQR 5–12). The most common presenting symptoms were dysuria (337/425, 79.3%), frequency (310/425, 72.9%) and suprapubic pain (272/425, 64.0%). Overall presenting symptoms were similar between individuals with STIs and UTIs and those with negative cultures (figure 2).

Antimicrobial treatment prior to clinic presentation was reported by 30 (7.1%), while the antibiotic bioassay was positive in 73 (17.4%) including 26 participants with HIV infection who were taking co-trimoxazole prophylaxis. Overall, 89 (20.9%) of the participants had either reported taking antibiotics or had a positive urine bioassay.

STI treatment was prescribed by the clinic nurses at the initial visit in 38 participants, of whom 20 (53%) had a positive STI test, while 63 (76%) of the 83 patients who had an STI were not prescribed STI treatment by the primary healthcare nurse.

A follow-up phone call was made with 393 (92.5%) of the participants. Most (334/393, 85.0%) reported that the symptoms had resolved or significantly improved since enrolment. Among the 59 participants whose symptoms did not improve or resolve, 15 had a positive urine culture and 6 had an STI, while 32 (54.2%) did not have any positive tests. Symptom resolution or improvement was reported by 70/76 (92.1%) participants with an STI, by 74/89 (83.1%) of those with a positive urine culture and by 127/159 (79.9%) with negative results in all tests. Among participants with positive urine cultures, symptoms improved or resolved in 65/69 (94.2%) of those

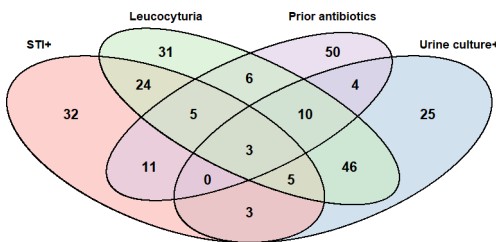

**Figure 1** Distribution of positive tests among study participants (170/425 samples; 40% were negative for all tests). STI, sexually transmitted infection.

**Table 2** Characteristics of the study participants according to the test result

| Characteristic | Total N=425 (%) | STI positive n=83 (%) | Urine culture positive n=96 (%) | Leucocyturia n=130 (%) | Prior antibiotics n=89 (%) | All tests negative n=170 (%) |
|---|---|---|---|---|---|---|
| Age (years) median (IQR) | 37.3 (26.8–48.0) | 28.4 (23.5–39.2) | 39.8 (26.0–50.0) | 34.2 (24.8–47.0) | 37.6 (27.6–49.4) | 39.4 (29.6–49.6) |
| Female sex | 275 (64.7) | 51 (61.5) | 73 (76.0) | 88 (67.7) | 51 (57.3) | 107 (62.9) |
| Completed at least secondary education | 359 (84.5) | 71 (85.5) | 75 (78.1) | 108 (83.1) | 77 (86.5) | 141 (82.9) |
| HIV-positive* | 172 (42.4) | 20 (25.6) | 42 (47.2) | 53 (43.4) | 38 (44.2) | 76 (46.1) |
| Pregnancy* | 35 (13.0) | 4 (7.8) | 7 (9.7) | 15 (17.2) | 5 (9.8) | 16 (15.5) |
| Married/have a partner | 325 (76.5) | 66 (79.5) | 74 (77.1) | 91 (70.0) | 67 (75.3) | 132 (77.7) |
| Duration of symptoms prior to presentation | 7 (5–12) | 7 (4–12) | 7 (4–11) | 7 (5–10) | 7 (5–12) | 9 (6–12) |
| Symptoms resolved on follow-up* | 334 (85.0) | 70 (92.1) | 74 (83.2) | 105 (89.0) | 73 (89.0) | 127 (79.9) |

*19 participants did not know their HIV status; 5 women did not know if they were pregnant; follow-up information available for 393 participants.
STI, sexually transmitted infection.

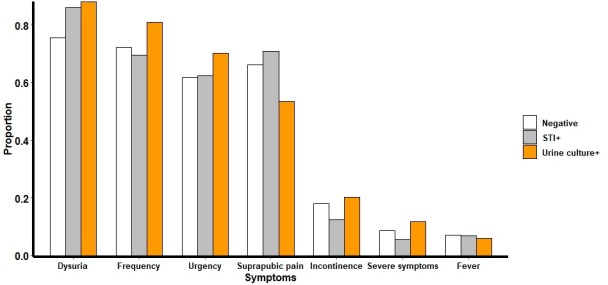

**Figure 2** Presenting symptoms in study participants according to urine culture and STI test result. Severe symptoms were defined as symptoms affecting daily activities. STI, sexually transmitted infection.

who reported having taken their prescribed treatment and in 9/20 (45%) of those who had not.

## DISCUSSION

This study found that only one in five people presenting to primary care in Harare with UTI symptoms had a positive urine culture with a uropathogen explaining their presenting symptoms. There was a high prevalence of STIs and evidence of prior antimicrobial use among study participants as possible explanations for the low proportion of positive urine cultures. However, in 40% of patients, symptoms remained unexplained.

Only 20% of study participants had a positive urine culture, which is similar to findings from studies conducted in other sub-Saharan African countries where culture positivity was found to be 11% and 36% among outpatients.[5 9] In contrast, studies conducted among women with cystitis in high-income countries, mostly from Europe and the Americas, report that 75%–80% of urine cultures reveal significant growth of uropathogens.[2 10 11] This may be explained by differences in access to healthcare, unregulated antibiotic use and prevalence of other conditions with similar symptoms between settings.[5 12–14]

In our study, the median time to presentation was 7 days, while studies from high-income settings report a much shorter time between symptom onset and clinic presentation of 2–3 days.[10 11] Thus, the lower yield of urine cultures in this study may be due to spontaneous resolution of infection because of the delays in accessing healthcare or self-treatment with antibiotics.[15] This may particularly be the case in patients with leucocyturia where symptoms can be explained by residual inflammation. Dysuria and leucocyturia in the absence of positive urine cultures may be due to other causes such as urogenital tuberculosis, schistosomiasis and other inflammatory conditions.[3] However, other infections are unlikely to be the cause of symptoms in our patient population, given that for the vast majority, the symptoms resolved or improved on follow-up and that haematuria was present in less than 1%.

Antimicrobial use prior to clinic presentation may also have contributed to a lower urine culture yield, with one in five study participants having evidence of taking

antibiotics prior to presentation to primary care. While only 7% of the study participants reported antimicrobial use in the previous 2 weeks, 17% had evidence of antimicrobial activity in their urine. Although antibiotic access from pharmacies is highly regulated in Zimbabwe, antibiotics can be acquired from the informal market.[16] Patients may not report on self-treatment at clinic presentation due to social desirability bias. Prior antimicrobial use was similar to that reported by patients from Senegal,[17] which had a culture positivity of 27% but lower than that in studies from the USA.[18] This suggests that asking patients about antimicrobials may not be sufficient in determining prior exposure as shown in findings from Ghana.[19] On the other hand, the antibiotic bioassay may overestimate antimicrobial use for treatment of infections by also detecting antimicrobials consumed unintentionally in food and water.[20] Alternative explanations for symptoms suggestive of UTIs such as dysuria include irritant hygienic products, spermicides and certain foods.[3]

STIs are highly prevalent in some sub-Saharan African settings,[21] and there is some symptom overlap between STIs and UTIs. A meta-analysis of studies from sub-Saharan Africa reported a pooled prevalence for *C. trachomatis* of 7.8% in women of reproductive age,[22] while a study from Zimbabwe testing mainly asymptomatic adolescents and young people found a prevalence of 17% of *C. trachomatis* and/or *N. gonorrhoeae*.[23] While STIs may be asymptomatic particularly in women, patients with symptoms due to STIs frequently experience diagnostic and treatment delays leading to prolonged symptoms prior to presentation.[24] One of the study eligibility criteria included recent symptom onset (within 2 weeks). This was aimed at increasing the likelihood of UTIs compared with STIs. Despite this, STI prevalence in our study was almost 20%. Importantly, 1 in 15 women and 1 in 8 men tested positive for *N. gonorrhoeae*. The high prevalence of STIs may also explain the large proportion of men enrolled into the study who are generally more likely to experience symptoms from STIs which are similar to symptoms of UTIs.[24] The study may have missed other STIs because testing was limited to three major pathogens but did not include other organisms that may cause urethritis and urinary tract symptoms such as *Mycoplasma genitalium*, leading to an underestimation of STI prevalence. Furthermore, STIs may have been missed because the study used mid-stream urine samples, while first-catch samples are recommended for the diagnosis of STIs for optimal yield on urine samples. Genital ulcers caused by herpes simplex and vaginitis due to *Candida spp.* may explain symptoms in some patients. Also, men were not tested for *T. vaginalis* as this is less frequent in men.[25] Potential participants who did not seek care because of not being able to afford the clinic fees would have been missed.

Only a quarter of participants who tested positive for an STI underwent STI treatment on the initial clinic presentation because the attending nurse did not suspect an STI. This highlights the difficulties in differentiating the two conditions on clinical grounds. Access to diagnostic testing at primary care level is thus urgently needed.

The prevalence of STIs is not generalisable to all primary healthcare attendees, as the participants were selected on the basis of STI/UTI symptoms.

## CONCLUSIONS

This is the first study exploring the causes for negative urine cultures among adults presenting with UTI symptoms to primary care in a low-income setting. The findings of this study highlight the high STI burden in this patient population. The overlap in symptoms in patients diagnosed with UTIs and STIs, respectively, results in underdiagnosis and undertreatment of STIs. Diagnostic testing is paramount to ensure appropriate treatment. Very few participants who had an STI received appropriate treatment on clinic presentation because STIs were not suspected, emphasising the missed opportunities for diagnosis and treatment. The high prevalence of prior antibiotic exposure in patients presenting to primary care should prompt an improvement of regulations around antibiotic access to prevent development of antimicrobial resistance.

**Author affiliations**
[1]Clinical Research Department, London School of Hygiene & Tropical Medicine, London, UK
[2]Biomedical Research and Training Institute, Harare, Zimbabwe
[3]Department of Paediatric Infectious Disease, St Mary's Hospital, London, UK
[4]Department of Health, City of Harare Health Services Department, Harare, Zimbabwe
[5]AIDS and TB Unit, Ministry of Health and Child Care, Harare, Zimbabwe
[6]Division of Infectious and Tropical Medicine, Medical Centre of the University of Munich, Munich, Germany

**Contributors** IDO and KK conceived the study idea and planned the study. RAF contributed to the study idea. PC, KPEM and AM provided support with the resources needed for the study. SY and RAF provided support with conducting the study. IDO and MC conducted the laboratory testing and the interpretation of the results. IDO and KK analysed and interpreted the data and drafted the initial manuscript. MC, SY, DM, MM, PC, KPEM, AM and RAF reviewed and provided feedback on the initial manuscript. All authors read and approved the final manuscript.

**Funding** IDO received funding though the Wellcome Trust Clinical PhD Programme awarded to the London School of Hygiene & Tropical Medicine (grant number 203905/Z/16/Z). The funders had no role in study design, data collection and interpretation, or the decision to submit the work for publication. The study was funded by UK aid from the UK government; the views expressed, however, do not necessarily reflect the UK government's official policies.

**Competing interests** None declared.

**Patient and public involvement** Patients and/or the public were not involved in the design, conduct, reporting or dissemination plans of this research.

**Patient consent for publication** Not required.

**Ethics approval** The study was approved by the London School of Hygiene and Tropical Medicine ethics committee (ref. 16424) and the Medical Research Council of Zimbabwe (MRCZ/A/2406). All study participants provided written informed consent.

**Provenance and peer review** Not commissioned; externally peer reviewed.

**Data availability statement** Data are available upon reasonable request. All data relevant to the study are included in the article. Data availability statement: the data

underlying this study can be obtained upon request by emailing the corresponding author.

**ORCID iDs**
Ioana D Olaru http://orcid.org/0000-0003-3392-9257
Shunmay Yeung http://orcid.org/0000-0002-0997-0850
Michael Marks http://orcid.org/0000-0002-7585-4743

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
