## [Reviewer comments · BMJ Open]

ARTICLE DETAILS

TITLE (PROVISIONAL)	Sexually transmitted infections and prior antibiotic use as important causes for negative urine cultures among adults presenting with urinary tract infection symptoms to primary care clinics in Zimbabwe, a cross-sectional study
AUTHORS	David; Marks, Michael; Chonzi, Prosper; Masunda, Kudzai PE; Machiha, Anna; Ferrand, Rashida; Kranzer, Katharina

VERSION 1 – REVIEW

REVIEWER	Heytens, Stefan University of Ghent, Department of Public Health and Primary Care
REVIEW RETURNED	19-Apr-2021

GENERAL COMMENTS	this is a very interesting study revealing an important health care problem specific for the sub-Saharan region. P5 line 31: selection bias? Can the lowest income groups in low income countries afford this user fee or is an essential part excluded of this type of primary health care? P5 line 53: what is the diameter used to be considered positive? P6 line 51: concerning fig 1: maybe it could be made more clear to mention the total (between brackets) of STI+, LE+, urine culture + en those who had prior antibiotics. General question about discussion: What is common management in Zimbabwe and in sub-Saharan Africa? Is there a (local) guideline? The European guidelines cannot be used as the results of urine analysis are completely different. Could this be an issue of the discussion or conclusion? Considering these results, what would be a reasonable management? Also when taking into account the probably limited resources. What to do with the 40 % negative results? How could these patients be best taken care of? Taking into account the underdiagnosis of STI in patients consulting with urinary complaints it could be rational to first test for STI and in a second step, when STI test is negative to further examine for UTI (which is in essence a self limiting disease). In doing so a cost saving could be made. In High-income countries urinary culture is not needed in women with suggested uncomplicated UTI, but as 40% of these patients do not have a pos STI nor a pos culture, in sub-Saharan Africa it could be
--

	desirable to perform a culture, to avoid overtreatment with antibiotics.
REVIEWER	Mun, Seok Inje University College of Medicine, Division of Infectious Diseases
REVIEW RETURNED	24-Apr-2021
GENERAL COMMENTS	The aim of the study was to find possible causes of negative urine cultures in patients presenting with symptoms of UTI to primary health clinics in Harare. The study included 425 participants with a median age of 37.3 years, of whom 275 (64.7%) were women and 172 (40.5%) were HIV-positive. Urine cultures were positive in 96 (22.6%) participants. Eighty-three (19.5%) participants had an STI. And those with evidence of prior antimicrobial use but no STI and positive urine culture were 56 participants. In 170 (40.0%) all of the tests that were performed were negative. # The frequency of leukocyturia was very low (30.6%). In participants without leukocyturia, positive urine cultures may not indicate that symptoms were due to UTI. Other causes may contribute to UTI symptoms. Furthermore, evidence of prior antimicrobial use does not mean that the UTI caused the symptoms in participants without positive other tests. These could underestimate the proportion of other causes except for UTI in UTI symptoms. # Symptom resolution or improvement was reported by 74/89 (83.1%) of participants with a positive urine culture. Among participants with a positive urine culture, symptoms improved or resolved in 64/67 (95.5%) of those who reported having taken their prescribed treatment and in 9/20 (45%) of those who had not. The numbers of patients in the two sentences do not correspond with each other. # In figure 2, severe symptoms were not clearly defined.

VERSION 1 – AUTHOR RESPONSE

REVIEWER 1

Comment #1

P5 line 31: selection bias? Can the lowest income groups in low income countries afford this user fee or is an essential part excluded of this type of primary health care?

Response to comment #1

This is an excellent point and we agree that clinic fees might have impacted on healthcare access which may have led to selection bias. This is now included as part of the limitations "*Potential participants who did not seek care because of not being able to afford the clinic fees would have been missed.*"

An alternative to evaluating the prevalence of STIs would have been to conduct the study at community-level. However, the parent-study was focused on the prevalence of antimicrobial resistance among patients presenting with symptoms of urinary tract infection and thus required a clinic-setting.

Comment #2

P5 line 53: what is the diameter used to be considered positive?

Response to comment #2

Because the discs did not contain any antimicrobials (they were filter paper discs onto which the urine samples from the participants were inoculated), any growth inhibition was considered relevant for the

presence of antimicrobial residues in the urine sample. For each batch of tests, we used positive and negative controls to ensure the quality of testing results. This was also further clarified in the manuscript text: *“Antibacterial activity was considered to be present in the sample if a growth inhibition zone of any size around the disc was observed following incubation for 24 hours at 37°C.”*

Comment #3

P6 line 51: concerning fig 1: maybe it could be made more clear to mention the total (between brackets) of STI+, LE+, urine culture + en those who had prior antibiotics.

Response to comment #3

This information was added to the manuscript: *“Figure 1 shows the distribution of positive test results [STI positive 83 (19.6%), urine culture positive 96 (22.6%), leukocyturia 130 (30.6%) and prior antibiotics 89 (20.9%).”*

Comment #4

What is common management in Zimbabwe and in sub-Saharan Africa? Is there a (local) guideline? The European guidelines cannot be used as the results of urine analysis are completely different. Could this be an issue of the discussion or conclusion? Considering these results, what would be a reasonable management? Also when taking into account the probably limited resources.

Response to comment #4

Prescriptions in primary care are done in accordance to the local guidelines for Zimbabwe (The Essential Medicines List of Zimbabwe - https://extranet.who.int/ncdccc/Data/ZWE_D1_7th%20edition%20EDLIZ%202015%20_%20Final%20Version%20with%20Signatures.docx). These include recommendations for the management of sexually transmitted and urinary tract infections. A clarification with regards to the guidelines used for prescribing decisions was added in the manuscript. *“Treatment for STIs was prescribed during the initial visit by the clinic nurse according to routine practice following the Zimbabwean national guidelines and independent of study procedures.”* Generally, STIs in Zimbabwe are treated using a syndromic approach because diagnostic testing is not available. Within this study, patients received routine care when they presented to clinic and were prescribed STI treatment if an STI was suspected by the clinic nurse. Patients who had a positive STI test and who were not prescribed appropriate treatment during the initial visit, were called back to the clinic for treatment. Given the high prevalence of STIs identified in this study, a reasonable approach would be to increase the availability of STI testing. This issue is discussed in the manuscript.

We also fully agree that using European guidelines would not have been appropriate in this context.

Comment #5

What to do with the 40 % negative results? How could these patients be best taken care of?

Response to comment #5

Although a high proportion of participants remained undiagnosed, for most (80%) symptoms were transitory and had resolved or improved at follow up. We hypothesize that in some of these patients, symptoms could have been explained by urinary tract infections and other STIs for which testing was not conducted. The delayed presentation (with a median of 7 days of symptoms until healthcare was sought) may have led to the spontaneous resolution of urinary tract infections and negative urine cultures. Potentially, increasing healthcare access by for example decreasing clinic fees would reduce the delay in seeking care and improve patient management.

Comment #6

Taking into account the underdiagnosis of STI in patients consulting with urinary complaints it could be rational to first test for STI and in a second step, when STI test is negative to further examine for UTI (which is in essence a self limiting disease). In doing so a cost saving could be made. In High-income countries urinary culture is not needed in women with suggested uncomplicated UTI, but as 40% of these patients do not have a pos STI nor a pos culture, in sub-Saharan Africa it could be desirable to perform a culture, to avoid overtreatment with antibiotics.

Response to comment #6

Because of the economic hardships in Zimbabwe and other low-resource settings, we do not think that a two-step testing approach would be ideal. Patients face great challenges in accessing care by having to pay out-of-pocket for consultation fees, prescribed medicines and transport to the clinic. Therefore, if multiple clinic visits were required, patients may not come back for further assessment after their initial clinic visit. Ideally, rapid point of care testing for STIs would identify patients requiring STI treatment, while in patients with negative STI tests and suggestive symptoms, an UTI can be

considered. Despite a high proportion of urine cultures being negative we would not recommend routine urine cultures. We are aware that a large proportion of UTIs are self-limiting and hence urine cultures in women presenting with symptoms of UTIs for the first time in primary care settings are not needed.

REVIEWER 2

Comment #1

The frequency of leukocyturia was very low (30.6%). In participants without leukocyturia, positive urine cultures may not indicate that symptoms were due to UTI. Other causes may contribute to UTI symptoms. Furthermore, evidence of prior antimicrobial use does not mean that the UTI caused the symptoms in participants without positive other tests. These could underestimate the proportion of other causes except for UTI in UTI symptoms.

Response to comment #1

Thank you, we fully agree that positive urine cultures in symptomatic patients with low bacterial loads and no leukocyturia might be difficult to interpret especially in the context of a high STI prevalence and frequent prior antibiotic use. We also fully agree that patients may have taken antibiotics for reasons other than their current symptoms although taking antibiotics for their symptoms would be the most likely explanation.

Comment #2

Symptom resolution or improvement was reported by 74/89 (83.1%) of participants with a positive urine culture. Among participants with a positive urine culture, symptoms improved or resolved in 64/67 (95.5%) of those who reported having taken their prescribed treatment and in 9/20 (45%) of those who had not.

The numbers of patients in the two sentences do not correspond with each other.

Response to comment #3

Thank you for pointing out our error. The numbers were corrected. *“Among participants with positive urine cultures, symptoms improved or resolved in 65/69 (94.2%) of those who reported having taken their prescribed treatment and in 9/20 (45%) of those who had not.”*

Comment #3

In figure 2, severe symptoms were not clearly defined.

Response to comment #3

Severe symptoms were defined as symptoms limiting daily activities. The definition was added in the figure legend.

VERSION 2 – REVIEW

REVIEWER	Heytens, Stefan University of Ghent, Department of Public Health and Primary Care
REVIEW RETURNED	23-Jul-2021
GENERAL COMMENTS	thank you for your adequate and high quality answers
REVIEWER	Mun, SEok Inje University College of Medicine, Division of Infectious Diseases
REVIEW RETURNED	14-Jul-2021
GENERAL COMMENTS	Summary: This study could be helpful in the diagnosis and treatment of patients with urinary tract infection symptoms at the primary care clinics in Zimbabwe. The prevalence of STI in patients with urinary tract infection symptoms was almost 20%. And in most patients with STI positive, STIs were not suspected. As the authors

	suggested, diagnostic testing for STI at the primary care level is needed in Zimbabwe. Discussion: 1. P9 line 48: I think that this sentence is not associated with reference 16.
--	---